# Factored Policy Gradients: Leveraging Structure for Efficient Learning in MOMDPs

**Thomas Spooner**
J. P. Morgan AI Research
thomas.spooner@jpmorgan.com

**Nelson Vadori**
J. P. Morgan AI Research
nelson.vadori@jpmorgan.com

**Sumitra Ganesh**
J. P. Morgan AI Research
sumitra.ganesh@jpmorgan.com

## Abstract

Policy gradient methods can solve complex tasks but often fail when the dimensionality of the action-space or objective multiplicity grow very large. This occurs, in part, because the variance on score-based gradient estimators scales quadratically. In this paper, we address this problem through a factor baseline which exploits independence structure encoded in a novel action-target influence network. Factored policy gradients (FPGs), which follow, provide a common framework for analysing key state-of-the-art algorithms, are shown to generalise traditional policy gradients, and yield a principled way of incorporating prior knowledge of a problem domain's generative processes. We provide an analysis of the proposed estimator and identify the conditions under which variance is reduced. The algorithmic aspects of FPGs are discussed, including optimal policy factorisation, as characterised by minimum biclique coverings, and the implications for the bias-variance trade-off of incorrectly specifying the network. Finally, we demonstrate the performance advantages of our algorithm on large-scale bandit and traffic intersection problems, providing a novel contribution to the latter in the form of a spatial approximation.

## 1 Introduction

Many sequential decision-making problems in the real-world have objectives that can be naturally decomposed into a set of conditionally independent targets. Control of water reservoirs, energy consumption optimisation, market making, cloud computing allocation, sewage flow systems, and robotics are but a few examples [36]. While many optimisation methods have been proposed [25, 34] — perhaps most prominently using Lagrangian scalarisation [46] — multi-agent learning has emerged as a promising new paradigm for sample-efficient learning [6]. In this class of algorithms, the multi-objective learning problem is cast into a centralised, co-operative stochastic game in which co-ordination is achieved through global coupling terms in each agent's objective/reward functions. For example, a grocer who must manage their stock could be decomposed into a collection of sub-agents that each manage a single type of produce, but are subject to a global constraint on inventory. This approach has been shown to be very effective in a number of domains [20, 50, 30, 24, 52], but presents both conceptual and technical issues.

The transformation of a multi-objective Markov decision process (MOMDP) [36] into a stochastic game is a non-trivial design challenge. In many cases there is no clear delineation between agents in the new system, nor an established way of performing the decomposition. What's more, it's unclear in many domains that a multi-agent perspective is appropriate, even as a technical trick. For example, the concurrent problems studied by Silver et al. [39] exhibit great levels of homogeneity,

lending themselves to the use of a shared policy which conditions on contextual information. The key challenge that we address in this paper is precisely how to scale these single-agent methods — specifically, policy gradients — in a *principled* way. As we shall see, this study reveals that existing methods in both single- and multi-agent multi-objective optimisation can be formulated as special cases of a wider family of algorithms we entitle *factored policy gradients*. The contributions of this paper are summarised below:

1. We introduce ***influence networks*** as a framework for modelling probabilistic relationships between actions and objectives in an MOMDP, and show how they can be combined with ***policy factorisation*** via graph partitioning.

2. We propose a new control variate — the ***factor baseline*** — that exploits independence structures within a (factored) influence network, and show how this gives rise to a novel class of algorithms to which we ascribe the name ***factored policy gradients***.

3. We show that FPGs ***generalise traditional policy gradient estimators*** and provide a common framework for analysing state-of-the-art algorithms in the literature including action-dependent baselines and counterfactual policy gradients.

4. The ***variance properties*** of our family of algorithms are studied, and ***minimum factorisation*** is put forward as a principled way of applying FPGs, with theoretical results around the existence and uniqueness of the characterisation.

5. The final contribution is to illustrate the effectiveness of our approach over traditional estimators on two ***high-dimensional benchmark domains***.

## 1.1 Related Work

**Policy gradients.** Variance reduction techniques in the context of policy gradient methods have been studied for some time. The seminal work of Konda and Tsitsiklis [19] was one of the earliest works that identified the use of a critic as beneficial for learning. Since then, baselines (or, control variates) have received much attention. In 2001, Weaver and Tao [53] presented the first formal analysis of their properties, and later Greensmith et al. [13] proved several key results around optimality. More recently, these techniques have been extended to include action-dependent baselines [47, 22, 12, 57, 9], though the source of their apparent success has been questioned by some [49] who suggest that subtle implementation details were the true driver. It has also been shown that one can reduce variance by better accounting for the structure of the action-space, such as bounds [4, 10] or more general topological properties [7]. The SVRPG approach of Papini et al. [29] also addresses variance concerns in policy gradients by leveraging advances in supervised learning, and the generalised advantage estimator of Schulman et al. [37] has been proposed as a method for reducing variance in actor-critic methods with fantastic empirical results; both of these can be combined with baselines and the techniques we present in this work. **Factorisation.** In a related, but distinct line of work, factorisation has been proposed to better leverage the transition structure of MDPs; see e.g. [2, 15, 44]. Indeed, the notion of causality has also been utilised in work by Jonsson and Barto [16]. Most recently, Oliehoek et al. [27] presented an elegant framework for harnessing the *influence* of other agents (from the perspective of self) in multi-agent systems. This approach is complementary to the work presented in this paper, and more recent extensions have significantly advanced the state-of-the-art [41, 5, 28]; we build upon these principles. There is also a long line of research on "influence diagrams" that is pertinent to this work. While the majority of this effort has been focused on dynamic programming, the ideas are very closely related to ours and indeed we see this work as a natural extension of these concepts [45]. **Miscellaneous.** Causal/graphical modelling has seen past applications in reinforcement learning [11]. Indeed, our proposed influence network is related to, but distinct from, the action influence models introduced by Madumal et al. [23] for explainability. There, the intention was to construct policies that can justify actions with respect to the observation space. Here, the intention was to exploit independence structure in MOMDPs for scalability and efficiency.

## 2 Background

A regular discrete-time Markov decision process (MDP) is a tuple $\mathcal{M} \doteq (\mathcal{S}, \mathcal{A}, \mathcal{R}, p, p_0)$, comprising: a *state space* $\mathcal{S}$, *action space* $\mathcal{A}$, and set of *rewards* $\mathcal{R} \subseteq \mathbb{R}$. The dynamics of the MDP are driven by an *initial state distribution* such that $s_0 \sim p_0(\cdot)$ and a stationary *transition kernel* where $(r_t, s_{t+1}) \sim$

$p(\cdot, \cdot \mid s_t, \boldsymbol{a}_t)$ satisfies the Markov property, $p(r_t, s_{t+1} \mid h_t) = p(r_t, s_{t+1} \mid s_t, \boldsymbol{a}_t)$, for any history $h_t \doteq (s_0, \boldsymbol{a}_0, r_0, s_1, \ldots, s_t, \boldsymbol{a}_t)$. Given an MDP, a (stochastic) policy, parameterised by $\boldsymbol{\theta} \in \mathbb{R}^n$, is a mapping $\pi_{\boldsymbol{\theta}} : \mathcal{S} \times \mathbb{R}^n \to \mathcal{P}(\mathcal{A})$ from states and weights to the set of probability measures on $\mathcal{A}$. The conditional probability density of an action $\boldsymbol{a}$ is denoted by $\pi_{\boldsymbol{\theta}}(\boldsymbol{a} \mid s) \doteq \mathbb{P}(\boldsymbol{a} \in \mathrm{d}\boldsymbol{a} \mid s, \boldsymbol{\theta})$ and we assume throughout that $\pi_{\boldsymbol{\theta}}$ is continuously differentiable with respect to $\boldsymbol{\theta}$. For a given policy, the *return* starting from time $t$ is defined as the discounted sum of future rewards, $G_t \doteq \sum_{k=0}^{T} \gamma^k r_{t+k+1}$, where $\gamma \in [0, 1]$ is the discount rate and $T$ is the terminal time [42]. *Value functions* express the expected value of returns generated from a given state or state-action pair under the MDP's transition dynamics and policy $\pi$: that is, $v_\pi(s) \doteq \mathbb{E}_\pi[G_t \mid s_t = s]$ and $q_\pi(s, \boldsymbol{a}) \doteq \mathbb{E}_\pi[G_t \mid s_t = s, \boldsymbol{a}_t = \boldsymbol{a}]$. The objective in *control* is to find a policy that maximises $v_\pi$ for all states with non-zero measure under $p_0$, denoted by the Lesbesgue integral $J(\boldsymbol{\theta}) \doteq \mathbb{E}_{p_0}[v_{\pi_{\boldsymbol{\theta}}}(s_0)] = \int_\mathcal{S} v_{\pi_{\boldsymbol{\theta}}}(s_0) \, \mathrm{d}p_0(s_0)$.

## 2.1 Policy Search

In this paper, we focus on policy gradient methods which optimise the parameters $\boldsymbol{\theta}$ directly. This is achieved, in general, by performing gradient ascent on $J(\boldsymbol{\theta})$, for which Sutton et al. [43] derived

$$\nabla_{\boldsymbol{\theta}} J(\boldsymbol{\theta}) = \mathbb{E}_{\pi_{\boldsymbol{\theta}}, \rho_{\pi_{\boldsymbol{\theta}}}}[(q_{\pi_{\boldsymbol{\theta}}}(s, \boldsymbol{a}) - b(s)) \, \boldsymbol{z}], \tag{1}$$

where $\boldsymbol{z} \doteq \nabla_{\boldsymbol{\theta}} \ln \pi_{\boldsymbol{\theta}}(\boldsymbol{a} \mid s)$ is the policy's score vector, $\rho_{\pi_{\boldsymbol{\theta}}}(s) \doteq \int_\mathcal{S} \sum_{t=0}^{\infty} \gamma^t p(s_t = s \mid \mathrm{d}s_0, \pi_{\boldsymbol{\theta}})$ denotes the (improper) discounted-ergodic occupancy measure, and $b(s)$ is a state-dependent baseline (or, control variate) [33]. Here, $p(s_t = s \mid s_0, \pi_{\boldsymbol{\theta}})$ is the probability of transitioning from $s_0 \to s$ in $t$ steps under $\pi_{\boldsymbol{\theta}}$. Equation 1 is convenient for a number of reasons: 1. it is a score-based estimator [26]; and 2. it falls under the class of stochastic approximation algorithms [1]. This is important as it means $q_\pi(s, \boldsymbol{a})$ may be replaced by *any* unbiased quantity, say $\psi : \mathcal{S} \times \mathcal{A} \to \mathbb{R}$, such that $\mathbb{E}_{\pi, \rho_\pi}[\psi(s, \boldsymbol{a})] = q_\pi(s, \boldsymbol{a})$, while retaining convergence guarantees. It also implies that optimisation can be performed using stochastic gradient estimates, the standard variant of which is defined below.

**Definition 2.1 (VPGs).** The *vanilla policy gradient* estimator for target-baseline pair $(\psi, b)$ is denoted

$$\boldsymbol{g}^{\mathrm{V}}(s, \boldsymbol{a}) \doteq [\psi(s, \boldsymbol{a}) - b(s)] \, \boldsymbol{z}, \tag{2}$$

where $\nabla_{\boldsymbol{\theta}} J(\boldsymbol{\theta}) = \mathbb{E}_{\pi_{\boldsymbol{\theta}}, \rho_{\pi_{\boldsymbol{\theta}}}}[\boldsymbol{g}^{\mathrm{V}}(s, \boldsymbol{a})]$.

## 2.2 Factored (Action-Space) MDPs

In this paper, we consider the class of MDPs in which the action-space factors into a product, $\mathcal{A} \doteq \bigotimes_{i=1}^n \mathcal{A}_i = \mathcal{A}_1 \times \cdots \times \mathcal{A}_n$, for some $n$. This is satisfied trivially when $n = 1$ and $\mathcal{A}_1 = \mathcal{A}$, but also holds in many common settings, such as $\mathcal{A} \doteq \mathbb{R}^n$, which factorises $n$ times as $\bigotimes_{i=1}^n \mathbb{R}$. This is equivalent to requiring that actions, $\boldsymbol{a} \in \mathcal{A}$, admit a "subscript" operation; without necessarily having $\mathcal{A}$ be a vector space. For example, one could have an action-space of the form $\mathcal{A} \doteq \mathbb{R} \times \mathbb{N}$ such that, for any $\boldsymbol{a} \in \mathcal{A}$, $a_1 \in \mathbb{R}$ and $a_2 \in \mathbb{N}$. To this end, we introduce the notion of partition maps which will feature throughout the paper.

**Definition 2.2 (Partition Map).** Define $\mathcal{X} \doteq \bigotimes_{i=1}^n \mathcal{X}_i$ and $J \subseteq [n]$ with $\mathcal{X}_J \doteq \bigotimes_{j \in J} \mathcal{X}_j$ such that a partition map (PM) for a pair $(\mathcal{X}, J)$ is a function $\sigma : \mathcal{X} \to \mathcal{X}_J$ with complement $\bar{\sigma} : \mathcal{X} \to \mathcal{X}_{[n] \setminus J}$.

Partition maps are an extension of the canonical projections of the product topology, and are equivalent to the scope operator used by Tian et al. [48]. For example, if $(a_1, a_2, a_3) \doteq \boldsymbol{a} \in \mathcal{A} \doteq \mathbb{R}^3$ denotes a three-dimensional real action-space, then one possible PM is given by $\sigma(\boldsymbol{a}) = (a_1, a_3)$ with complement $\bar{\sigma}(\boldsymbol{a}) = (a_2)$. Note that there should always exist a unique inverse operation that recovers the original space; in this case, it would be expressed as $f((a_1, a_3), (a_2)) = (a_1, a_2, a_3)$.

## 3 Influence Networks

Consider an MOMDP with scalarised objective given by

$$J(\boldsymbol{\theta}) \doteq \mathbb{E}_{p_0}\left[\psi(s, \boldsymbol{a}) \doteq \sum_{j=1}^m \lambda_j \psi_j(s, \sigma_j(\boldsymbol{a}))\right], \tag{3}$$

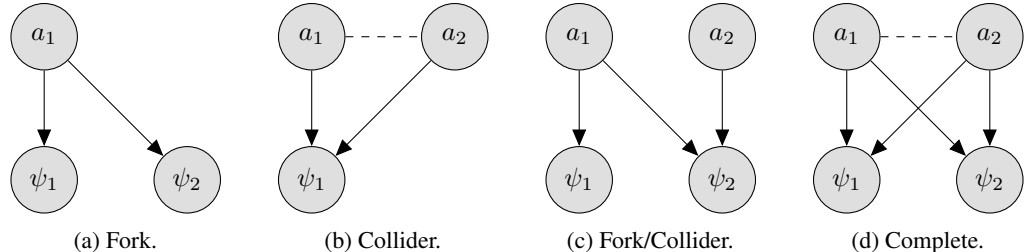

| (a) Fork. | (b) Collider. | (c) Fork/Collider. | (d) Complete. |

Figure 1: Influence network prototypes and action-target junction patterns [31, 32]. Edges depict dependencies between factors $a_i \in \mathcal{A}$ and targets $\psi_j \in \Psi$; and dashed lines a partition induced by the minimum factorisation.

where $\lambda_j \in \mathbb{R}$ for all $1 \leq j \leq m$ and each $\psi_j(s, \sigma_j(\boldsymbol{a}))$ denotes some target that depends on a single partition of the action components. Traditional MDPs can be seen as a special case in which $m = 1$, and $\psi = \psi_1 \doteq q_\pi$. The vector $\boldsymbol{\psi}(s, \boldsymbol{a})$ comprises the concatenation of all $m$ targets and each partitioning is dictated by the *non-empty* maps $\sigma_j(\boldsymbol{a})$, the form of which is intrinsic to the MOMDP. For convenience, let us denote the collection of targets comprising $\psi(s, \boldsymbol{a})$ by

$$\Psi \doteq \{\psi_j : \psi(s, \boldsymbol{a}) = \langle \boldsymbol{\lambda}, \boldsymbol{\psi}(s, \boldsymbol{a}) \rangle \}. \tag{4}$$

The intuition behind FPGs is derived from the observation that each factor of the action-space only *influences* a subset of the $m$ targets. Take, for example, Figure 1c which depicts an instance of an influence network between a 2-dimensional action vector and a 2-dimensional target. The edges suggest that $a_1$ affects the value of both $\psi_1$ and $\psi_2$, whereas $a_2$ only affects $\psi_2$. This corresponds to an objective of the form $\lambda_1 \psi_1(s, a_1) + \lambda_2 \psi_2(s, \boldsymbol{a})$, where each goal's domain derives from the edges of the graph. This is formalised in Definition 3.1 below.

**Definition 3.1** (**Influence Network**). A bipartite graph $\mathcal{G}(\mathcal{M}, \Psi) \doteq (I_\mathcal{A}, I_\Psi, E)$ is said to be the influence network of an MDP $\mathcal{M}$ and target set $\Psi$ if for $I_\mathcal{A} \doteq [|\mathcal{A}|]$ and $I_\Psi \doteq [|\Psi|]$, the presence of an edge, $e \in E$, between nodes $i \in I_\mathcal{A}$ and $j \in I_\Psi$ defines a causal relationship between the $i^{\text{th}}$ factor of $\mathcal{A}$ and the $j^{\text{th}}$ target $\psi_j(s, \sigma_j(\boldsymbol{a}))$.

An influence network can be seen as a structural equation model [31] in which each vertex in $I_\mathcal{A}$ has a single, unique parent which is exogenous and drives the randomness in action sampling, and each vertex in $I_\Psi$ has parents only in the set $I_\mathcal{A}$ as defined by the set of edges $E$. The structural equations along each edge $(i, j) \in E$ are given by the target functions themselves and the partition maps $\sigma_j$ mirror the parents of each node $j$. Some examples of influence networks are illustrated in Figure 1; see also the appendix. We now define the key concept of influence matrices.

**Definition 3.2** (**Influence Matrix**). Let $\boldsymbol{K}_\mathcal{G}$ denote the *biadjacency matrix* of an influence network $\mathcal{G}$, defined as the $|I_\mathcal{A}| \times |I_\Psi|$ boolean matrix with $K_{ij} = 1 \iff (i, j) \in E$ for $i \in I_\mathcal{A}$ and $j \in I_\Psi$.

Together, these definitions form a calculus for expressing the relationships between the factors of an action-space and the targets of an objective of the form in Equation 3. We remark that, from an algorithmic perspective, we are free to choose between two representations: graph-based, or partition map-based. The duality between $\mathcal{G}$ and $\boldsymbol{K}$, and the set $\{\sigma_j : j \in I_\Psi\}$, is intrinsic to our choice of notation and serves as a useful correspondence during analysis.

### 3.1 Policy Factorisation

Influence networks capture the relationships between $\mathcal{A}$ and $\Psi$, but policies are typically defined over groups of actions rather than the individual axes of $\mathcal{A}$. Consider, for example, a multi-asset trading problem in which an agent must quote buy and sell prices for each of $n$ distinct assets [14, 40]. There is a natural partitioning between each pair of prices and the $n$ sources of profit/loss, and one might therefore define the policy as a product of $n$ bivariate distributions as opposed to a full joint, or fully factored model. This choice over *policy factorisation* relates to the independence assumptions we make on the distribution $\pi_{\boldsymbol{\theta}}$ for the sake of performance. Indeed, in the majority of the literature, policies are defined using an isotropic distribution [57] since there is no domain knowledge to motivate more complex covariance structure. We formalise this below.

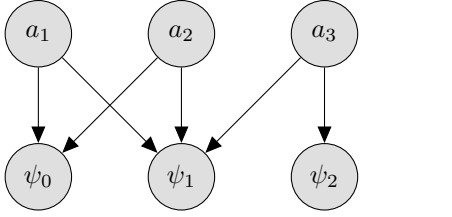 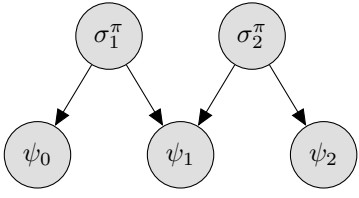

(a) Original Influence Network, $\mathcal{G}$.      (b) Factored Influence Network, $\mathcal{G}_\Sigma$.

Figure 2: Influence network transformation under a $\Sigma$-factorisation with $\sigma_1^\pi(\boldsymbol{a}) \doteq (a_1, a_2)$ and $\sigma_2^\pi(\boldsymbol{a}) \doteq (a_3)$. Here, $\Sigma$ corresponds to a minimum factorisation of the policy; i.e. $\Sigma = \Sigma^\star$.

**Definition 3.3** (**Policy Factorisation**). An $n$-fold *policy factorisation*, $\Sigma \doteq \{\sigma_i^\pi : i \in [n]\}$, is a set of disjoint partition maps that form a complete partitioning over the action space.

The definition above provides a means of expressing *any* joint policy distribution in terms of PMs,

$$\pi_{\boldsymbol{\theta}}(\boldsymbol{a} \,|\, s) \doteq \prod_{i=1}^n \pi_{i,\boldsymbol{\theta}}(\sigma_i^\pi(\boldsymbol{a}) \,|\, s), \tag{5}$$

where $\sigma_i^\pi \in \Sigma$ and $n = |\Sigma|$. This corresponds to a transformation of the underlying influence network where the action vertices are grouped under the $n$ policy factors and, for any $i, j \in [n]$, $i \neq j$, we have mutual independence: $\sigma_i^\pi(\boldsymbol{a}) \perp\!\!\!\perp \sigma_j^\pi(\boldsymbol{a})$. This is captured in the following concept.

**Definition 3.4** (**Factored Influence Network**). For a given influence network $\mathcal{G}$ and policy factorisation $\Sigma$, we define a *factored influence network*, $\mathcal{G}_\Sigma$, by replacing $I_\mathcal{A}$ with $I_\Sigma$, the set of partitioned vertices, and merge the corresponding edges to give $E_\Sigma$. Similarly, denote by $\boldsymbol{K}_\Sigma$ the influence matrix with respect to the $\Sigma$-factorisation.

Factored influence networks ascribe links between the policy factors in Equation 5 and the targets $\psi_j \in \Psi$. They play an important role in Section 4 and provide a refinement of Definition 3.1 which allows us to design more efficient algorithms. As an example, Figure 2 shows how one possible policy factorisation transforms an influence network $\mathcal{G}$ into $\mathcal{G}_\Sigma$. Note that while the action nodes and edges have been partitioned into policy factors, the fundamental topology with respect to the attribution of influence remains unchanged; i.e. no dependencies are lost.

## 4  Factored Policy Gradients

*Factored policy gradients* exploit factored influence networks by attributing each $\psi_j \in \Psi$ only to the policy factors that were probabilistically responsible for generating it; that is, those with a connecting edge in the given $\mathcal{G}_\Sigma$. The intuition is that the extraneous targets in the objective do not contribute to learning, but do contribute towards variance. For example, it would be counter-intuitive to include $\psi_2$ of Figure 2b in the update for $\pi_1$ since it played no generative role. Naturally, by removing these terms from the gradient estimator, we can improve the signal to noise ratio and yield more stable algorithms. This idea can be formulated into a set of baselines which are defined and validated below.

**Definition 4.1** (**Factor Baselines**). For a given $\mathcal{G}_\Sigma$, the *factor baselines* (FBs) are defined as

$$b_i^{\mathrm{F}}(s, \bar{\sigma}_i^\pi(\boldsymbol{a})) \doteq [(\mathbf{1} - \boldsymbol{K}_\Sigma)\,\boldsymbol{\lambda} \circ \boldsymbol{\psi}(s, \boldsymbol{a})]_i, \tag{6}$$

for all $i \in [|\Sigma|]$, where $\circ$ denotes the Hadamard product and $\mathbf{1}$ is to be taken as an all-ones matrix.

**Lemma 4.1.** *FBs are valid control variates if $\mathcal{G}_\Sigma$ is true to the MDP (i.e. unbiased).*

*Factor baselines* are related to the action-dependent baselines studied by Wu et al. [57] and Tucker et al. [49], as well as the methods employed by COMA [9] and DRPGs [3] in multi-agent systems. Note, however, that FBs are distinct in two key ways: 1. they adhere to the structure of the influence network and account not only for policy factorisation, but also the target multiplicity of MOMDPs; and 2. unlike past work, factor baselines were defined using an ansatz based on the structure implied by a given $\mathcal{G}_\Sigma$ as opposed to explicitly deriving the $\arg\min$ of the variance, or approximation thereof; see the appendix. This means that, unlike optimal baselines, FBs can be computed efficiently and

thus yield practical algorithms. Indeed, this very fact is why the state-value function is used so ubiquitously in traditional actor-critic methods as a state-dependent control variate despite being sub-optimal. It follows that we can define an analogous family of methods for MOMDPs with zero computational overhead.

**Proposition 1** (FPGs). *Take a $\Sigma$-factored policy $\pi_{\boldsymbol{\theta}}(\boldsymbol{a}|s)$ and $|\boldsymbol{\theta}| \times |\Sigma|$ matrix of scores $\boldsymbol{S}(s, \boldsymbol{a})$. Then, for target vector $\boldsymbol{\psi}(s, \boldsymbol{a})$ and multipliers $\boldsymbol{\lambda}$, the FPG estimator*

$$\boldsymbol{g}^{\mathrm{F}}(s, \boldsymbol{a}) \doteq \boldsymbol{S}(s, \boldsymbol{a}) \, \boldsymbol{K}_{\Sigma} \, \boldsymbol{\lambda} \circ \boldsymbol{\psi}(s, \boldsymbol{a}) \,, \tag{7}$$

*is an unbiased estimator of the true policy gradient; i.e. $\nabla_{\boldsymbol{\theta}} J(\boldsymbol{\theta}) = \mathbb{E}_{\pi_{\boldsymbol{\theta}}, \rho_{\pi_{\boldsymbol{\theta}}}} \big[ \boldsymbol{g}^{\mathrm{F}}(s, \boldsymbol{a}) \big].$*

Proposition 1 above shows that the VPG estimator given in Definition 2.1 can be expressed in our calculus as $\boldsymbol{S}\boldsymbol{1}\boldsymbol{\lambda} \circ \boldsymbol{\psi}$, where $\boldsymbol{1}$ is an all-ones matrix and, traditionally, $\psi \doteq q_{\pi}$; note that one can still include other baselines in Equation 7 such as $v_{\pi}$. In other words, *Proposition 1 strictly generalises the policy gradient theorem* [43] and, by virtue of it's unbiasedness, thus retains all convergence guarantees. We also see that both COMA [9] and DRPGs [3] are special cases in which the influence network reflects the separation of agents with $\boldsymbol{K}_{\Sigma}$ a square, and often diagonal matrix.

## 4.1 Variance Analysis

The variance reducing effect of FBs comprises two terms: 1. a quadratic and thus non-negative component which scales with the *second moments* of $b_i^{\mathrm{F}}$; and 2. a linear term which scales with the *expected values* of $b_i^{\mathrm{F}}$. This is shown in the following result.

**Proposition 2** (Variance Decomposition). *Let $\boldsymbol{g}_i$ denote a gradient estimate for the $i^{th}$ factor of a $\Sigma$-factored policy $\pi_{\boldsymbol{\theta}}$ (Equation 5). Then, $\Delta\mathbb{V}_i \doteq \mathbb{V}\big[\boldsymbol{g}_i^{\mathrm{V}}\big] - \mathbb{V}\big[\boldsymbol{g}_i^{\mathrm{F}}\big]$, satisfies*

$$\Delta\mathbb{V}_i = \alpha_i \, \mathbb{E}_{\bar{\sigma}_i^{\pi}(\boldsymbol{a})}\Big[\big(b_i^{\mathrm{F}}\big)^2\Big] + 2\beta_i \mathbb{E}_{\bar{\sigma}_i^{\pi}(\boldsymbol{a})}\big[b_i^{\mathrm{F}}\big] \,, \tag{8}$$

*where $\boldsymbol{z}_i \doteq \nabla_{\boldsymbol{\theta}} \ln \pi_{i,\boldsymbol{\theta}}(\boldsymbol{a} \,|\, s)$, $\alpha_i \doteq \mathbb{E}_{\sigma_i^{\pi}(\boldsymbol{a})}[\langle \boldsymbol{z}_i, \boldsymbol{z}_i \rangle] \geq 0$ and $\beta_i \doteq \mathbb{E}_{\sigma_i^{\pi}(\boldsymbol{a})}\big[\langle \boldsymbol{z}_i, \boldsymbol{z}_i \rangle \big(\psi + b_i^{\mathrm{F}}\big)\big].$*

The first of these two terms is a "free lunch" which removes the targets that are not probabilistically related to each factor. The linear term, on the other hand, couples the adjusted target with the entries that were removed by the baseline. This suggests that asymmetry and covariance can have a regularising effect in VPGs that is not present in FPGs — a manifestation of the properties of control variates [26]. Now, if we do not assume that the target functions are bounded, then the linear term in Equation 8 can grow arbitrarily in either direction, but we typically require that rewards are restricted to some compact subset $\mathcal{R} \subset \mathbb{R}$ to avoid this. Below, we show that if a similar requirement holds for each target function — namely, that $\inf_{\mathcal{S}, \mathcal{A}} \psi_j$ is well defined for each $\psi_j \in \Psi$ — then we can always construct a set of mappings that constrain (8) to be non-negative without biasing the gradient.

**Corollary 4.1** (Non-Negative Variance Reduction). *Let $\psi(s, \boldsymbol{a})$ be of the form in Equation 3. If $\psi_j(s, \boldsymbol{a}) \geq \underline{\psi_j}$ for all $(s, \boldsymbol{a}) \in \mathcal{S} \times \mathcal{A}$ and $j \in [m]$, with $|\underline{\psi_j}| < \infty$, then there exists a linear translation, $\psi_i \to \psi_i - \sum_{j=1}^m \lambda_j \underline{\psi_j}$, which leaves the gradient unbiased but yields $\Delta\mathbb{V}_i \geq 0$.*[1]

Interestingly, numerical experiments on a pair of continuum armed bandits suggest that this transformation is seldom necessary; see Figure 3. As the number of policy factors and targets grow, so too does the potential discrepancy in magnitude between the quadratic and linear terms in Equation 8. The former starts to dominate even for small $|\Sigma|$. This is particularly prevalent when the influence matrix $\boldsymbol{K}_{\Sigma}$ is very sparse and the baselines have wide coverage over $\Psi$. In other words, applying FBs when the influence network is very dense or even complete will not yield tangible benefits (e.g. in Atari games), but applying them to a problem with a rich structure, such as traffic networks, will almost certainly yield a significant net reduction in variance.

**Bias-Variance Trade-Off.** It is important to note that, in real-world problems, one does not always know the exact structure of the influence network underling an MDP ex ante. This poses a challenge since incorrectly *removing* edges can introduce bias and thus constrain the space of solutions that can be found by FPGs. Note, however, that this may not always be a problem, since a small amount

---

[1]This inequality can be made strict if either $\alpha_i > 0$ or $\beta_i > 0$ — where the former equates to having a non-zero trace of the Fisher information matrix — and a small $\varepsilon > 0$ is added to the translation.

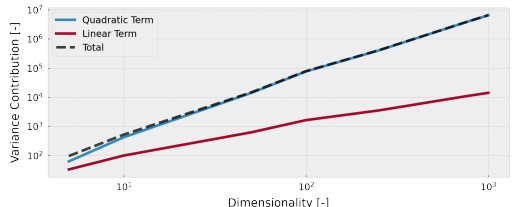
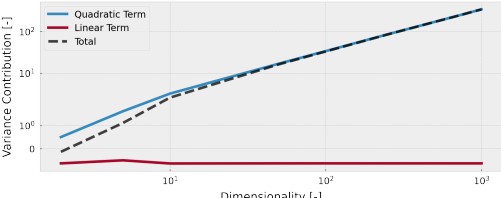

(a) **Search bandit**: $\text{Cost}(\boldsymbol{a}) \doteq -\|\boldsymbol{a} - \boldsymbol{c}\|_1$ with fixed centroid vector $\boldsymbol{c} \in \mathbb{R}^n$.

(b) **ReLU bandit**: $\text{Cost}(\boldsymbol{a}) \doteq -\sum_{i=1}^n \max(e_i a_i, 0)$ with fixed sign vector $\boldsymbol{e} \in \{-1, 1\}^n$.

Figure 3: Variance decomposition on a symmetric log scale for two bandit problems as a function of action-space dimensionality. Each term was computed using Monte-Carlo estimation with $10^5$ samples and taking the arithmetic mean across all policy factors.

bias for a large reduction in variance can be desirable. Furthermore, one could leverage curriculum learning to train the policy on (presumed) influence networks with increasing connectedness over time. This trade-off between bias and variance is present in many machine learning settings, and depends strongly on the problem at hand; we explore this empirically in Section 5.2.

## 4.2 Minimum Factorisation

For many classes of fully-observable MDPs, any policy factorisation is theoretically viable: we can fully factor the policy such that each action dimension is independent of all others; or, at the other extreme, treat the policy as a full joint distribution over $\mathcal{A}$. This holds because, in many classes of (fully-observable) MDPs, there exists at least one deterministic optimal policy [55, 35]. The covariance acts as a driver of exploration, and it's initial value only affects the rate of convergence.[2] As a result, most research uses an isotropic Gaussian with diagonal covariance to avoid the cost of matrix inversion. This poses an interesting question: is there an "optimal" policy factorisation, $\Sigma_{\mathcal{G}}^\star$, associated with an influence network $\mathcal{G}$? Below we offer a possible characterisation.

**Definition 4.2** (Minimum Factorisation). A minimum factorisation (MF), $\Sigma_{\mathcal{G}}^\star$, of an influence network, $\mathcal{G}$, is the *minimum biclique vertex cover, disjoint amongst* $I_{\mathcal{A}}$.

It follows from Definition 4.2 that for any $\Sigma_{\mathcal{G}}^\star$, each $\sigma_i \in \Sigma_{\mathcal{G}}^\star$ is a biclique (i.e. complete bipartite subgraph) of the original influence network $\mathcal{G}$, and that the bipartite dimension is equal to the number of policy factors. For example, one can trivially verify that Figure 2b is an MF of the original graph; see also the reductions in Figure 1. In essence, an MF describes a complete partitioning over action vertices — so as to define a proper distribution — where each group is a biclique with the same set of outgoing edges. The "minimum" qualifier then ensures that the maximum number of nodes are included in each of these groups, a property which allows us to prove the following result:

**Theorem 4.1.** *The MF* $\Sigma_{\mathcal{G}}^\star$ *always exists and is unique.*

Minimum factorisation is a natural construction for the problem domains studied in this paper; see Section 5. It also yields factored policies which, generally, expose the minimum infimum bound on variance for a given influence network. This follows from the fact that an MF yields the greatest freedom to express covariance structure within each of the policy factors whilst also maximising the quadratic term in Equation 8. In fact, when each action corresponds to a single unique target, the MF enjoys a lower bound on variance that is linear in the number of factors. Finally, we remark that, whilst closely related to vertex covering problems (which are known to be NP-complete [18]), we observed experimentally that finding the MF can be done trivially in polynomial time; see e.g. [8].

---

[2]Note that this is not true in general: the policy's covariance structure impacts the set of reachable solutions in partially-observable MDPs and stochastic games, for example.

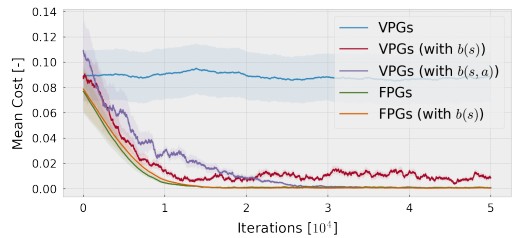 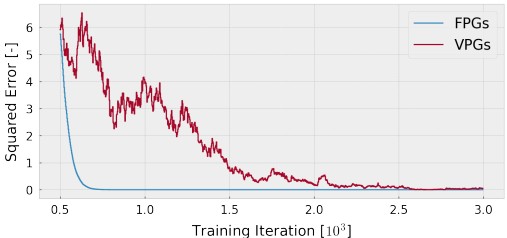

(a) Convergence of CPGs (with and w/o additional state-dependent baseline) compared with VPGs using different baselines. The error bands denote the standard error on the mean over 10 random seeds.

(b) Squared error between $a_n$ and $c_n$ during learning for the search bandit with $k = n - 1$.

Figure 4: Performance analysis of CPGs on the search bandit domain.

Table 1: Empirical wall-clock estimates for the time-complexity (iterations per second) of VPGs and FPGs, with and without additional baselines. For each algorithm, the mean and sample standard deviation were computed across the 10 random seeds used to generate Figure 4a.

| Method | Baseline | Mean [it / s] | Std Dev [it / s] |
|---|---|---|---|
| VPGs | - | 10534 | 87 |
|  | $b(s)$ | 9885 | 81 |
|  | $b(s, a)$ | 80 | 1 |
| FPGs | - | 9950 | 157 |
|  | $b(s)$ | 9670 | 126 |

## 5 Numerical Experiments

### 5.1 Search Bandits

Consider an $n \doteq 1000$ dimensional continuum armed bandit with action space in $\mathbb{R}^n$, and cost function: $\text{Cost}(\boldsymbol{a}) \doteq ||\boldsymbol{a} - \boldsymbol{c}||_1 + \lambda \zeta(\boldsymbol{a})$, where $\boldsymbol{c} \in \mathbb{R}^n$, $\lambda \geq 0$ and $\zeta : \mathcal{A} \to \mathbb{R}_+$ is a penalty function. This describes a search problem in which the agent must locate the centroid $\boldsymbol{c}$ subject to an action-regularisation penalty. It abstracts away the prediction aspects of MDP settings, and allows us to focus only on scalability; note that this problem is closely related to the bandit studied by Wu et al. [57] for the same purpose. In our experiments, the centroids were initialised with a uniform distribution, $\boldsymbol{c} \sim \mathcal{U}(-5, 5)$ and were held fixed between episodes. The policy was defined as an isotropic Gaussian with fixed covariance, $\text{diag}(\mathbf{1})$, and initial location vector $\boldsymbol{\mu} \doteq \mathbf{0}$. The influence network was specified such that each policy factor, $\pi_{i,\boldsymbol{\theta}}$ for $i \in [n]$, used a reward target $\psi_i(\boldsymbol{a}) \doteq -\Delta_i(a_i) - \lambda \zeta(a_i)$, with $\Delta_i(a) \doteq |a - c_i|$, amounting to a collection of $n$ forks (Figure 1a). The parameter vector, $\boldsymbol{\mu}$, was updated at each time step, and the hyperparameters are provided in the appendix.

We began by examining the case where $\lambda = 0$ and the co-ordinate axes were fully decoupled. For VPGs, we note that stability was only possible without a baseline if an extremely low learning rate was used; see the appendix. Including a baseline dramatically improved performance, with the action-dependent case, $b(s, a)$, also leading to better asymptotic behaviour at the expense of a two orders of magnitude longer train-time according to the wall-clock compared with all other algorithms (VPGs and FPGs); see Table 1. In comparison, FPGs, both with and without a learnt state-dependent baseline, yielded significantly reduced variance, leading to faster learning, more consistent convergence and highly robust asymptotic stability.

We then studied the impact of coupling terms in the cost function; i.e. $\lambda > 0$. For this, we considered a family of penalties taking the form of partially applied $\ell_2$ norms: $\zeta_k(\boldsymbol{a}) \doteq \sqrt{\sum_{i=1}^{k} a_i^2}$, with $1 \leq k \leq n$. This set of functions allowed us to vary the penalty attribution across the $n$ factors of $\mathcal{A}$. Further examples demonstrating the performance advantage of FPGs — for $k = n$ and $k = n/2$

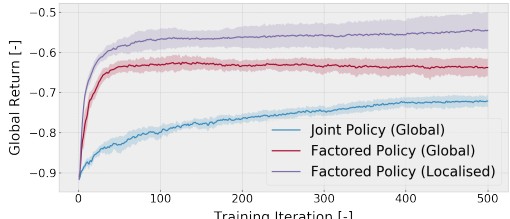
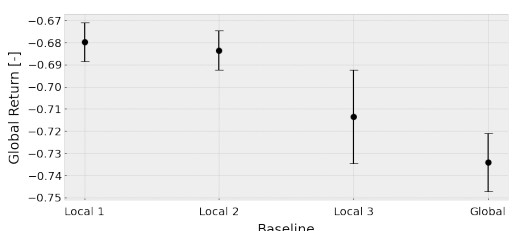
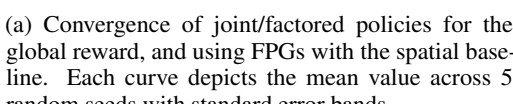

(a) Convergence of joint/factored policies for the global reward, and using FPGs with the spatial baseline. Each curve depicts the mean value across 5 random seeds with standard error bands.

(b) Performance degradation as a function of the $n$-level spatial baseline approximation in a $2 \times 6$ grid network. Each point is the average terminal value across 5 seeds and with standard error bands.

Figure 5: Performance analysis of FPGs (with PPO and GAE) on the traffic network domain.

— are given in the appendix. In both cases, the improvement due to FB adjustments was found to be non-negative for every combination of learning rate and action space. This confirms that FPGs can indeed handle coupled targets and retains the variance reduction benefits that were explored in Section 4.1. As an illustrative example, consider the case where $k = n - 1$ and all but the last action dimension are subject to a penalty. This is a particularly challenging setting for VPGs because the magnitude of the combined cost function is much greater than $\Delta_n(a_n)$, leading to an *aliasing* of the final component of the action vector in the gradient. The result, as exemplified in Figure 4b, was that VPGs favoured reduction of overall error, and was therefore exposed to poor per-dimension performance; hence the increased noise in the $a_n$ error process. FPGs avoid this effect by attributing gradients directly.

## 5.2 Traffic Networks

We now consider a classic traffic optimisation domain to demonstrate the scalability of our approach to large real-world problems. In particular, we consider variants of the $(3 \times 3)$ grid network benchmark environment — as originally proposed by Vinitsky et al. [51] — that is provided by the outstanding `Flow` framework [56, 21]. In this setting, the RL agent is challenged with managing a set of traffic lights with the objective of minimising the delay to vehicles travelling in the network; the configuration should be taken as identical unless explicitly stated. This requires a significant level of co-ordination, and indeed multi-agent approaches have shown exemplary performance in this space [51, 54]. However, much as with the search bandit, the probability of aliasing effects increases substantially with the number of lights/intersections; i.e. the dimensionality of the action-space. This affects both single- and multi-agent approaches when the global reward is used to optimise the policy.

To this end, we propose a "baseline" that *removes reward terms derived from streets/edges that are not directly connected to a given traffic light*. This is based on the hypothesis that the local problem is sufficiently isolated from the rest of the system that we may still find a (near)-optimal solution; much as with local-form models [27]. Of course, this could introduce bias at the cost of variance if we are incorrect (see Section 4.1), but this turns out to be an effective trade-off as exemplified in Figure 5a.[3] In this plot we compare the performance of three policies learnt using PPO [38] and GAE [37] (with an additional state-dependent baseline): (1) a naïve joint policy over the 9-dimensional action-space trained against the global reward; (2) a shared policy trained on the global reward; and (3) a shared policy using the local spatial baseline. In methods 2 and 3, a shared policy refers to the use of a single univariate policy across all nine traffic lights, where only local information and identity variables are provided. The global reward in this case was defined as the negative of the mean delay introduced in the system minus a scaled penalty on vehicles at near standstill; see the appendix for more details.

As expected, we observe that the FBs improve learning efficiency, but, perhaps surprisingly, we also find that the *asymptotic behaviour is also superior*. We posit that this relates to the fact that, with a fixed learning rate, stochastic gradient descent cannot distinguish between points within a ball of the optimum solution with radius that scales with the variance on the estimator. In other words, significant reductions in variance, even if they introduce a small amount of bias, may increase the likelihood of reaching the true optimal solution by virtue of having much greater precision.

---

[3]Note that the standard errors may slightly underestimate the population level due to the low sample size.

To better understand this trade-off, we also explored the impact of "expanding" the local baseline in a larger system of $2 \times 6$ intersections. With this new baseline we retain reward terms derived from lights up to $n$ edges away in either the east or west directions. The variable $n$ thus provides a dial to directly tweak the bias and variance of the policy gradient estimator (i.e. increasing $n$ reduces bias but increases variance). The result, as shown in Figure 5b, suggest that performance decreases monotonically as a function of $n$. This corroborates the claim in Section 4.1 that introducing some bias in exchange for a reduction in variance can be a worthwhile trade-off in large problems.

## 6    Conclusion

Factored policy gradients derive from the observation that many MOMDPs exhibit redundancy in their reward structure. Here, we have characterised this phenomenon using graphical modelling, and demonstrated that conditional independence between factors of the action-space and the optimisation targets can be exploited. The resulting family of algorithms subsume many existing approaches in the literature. Our results in large-scale bandit and concurrent traffic management problems suggest that FPGs are highly suited to real-world problems, and may provide a way of scaling RL to domains that have hitherto remained intractable. What's more, FPGs are compatible with other techniques that improve policy gradient performance. For example, they can be extended to use natural gradients by pre-multiplying Equation 7 by the inverse Fisher information matrix [17], and can even use additional baselines to reduce variance even further, as in Section 5.2. In future work we intend to address the following interesting questions: (a) Can we infer/adapt the structure of influence networks? (b) Are there canonical structures within $S$ and $K$? (c) What theoretical insights can be derived from a more detailed analysis of the variance properties of FPGs? We argue that factored approaches such as FPGs — which are complementary to ideas like influence-based abstraction [27] — are a promising direction for practical RL. Addressing some of these questions, we believe, would thus be of great value to the community.

## Acknowledgments and Disclosure of Funding

The authors would like to acknowledge our colleagues Joshua Lockhart, Jason Long and Rui Silva for their input and suggestions at various key stages of the research. This work was conducted by JPMorgan's AI Research group which has no external funding sources; i.e. it was a self-funded project. No sources of financial competing interests (or otherwise) are attributed to this line of research.

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
