## A Factor Baselines

As shown in the main text, under the assumption that the influence network is unbiased, our factor baselines are indeed valid control variates. We prove this result below, repeating the statement itself for posterity and providing a supplementary lemma on control variates as a restatement of known results.

**Lemma A.1** (**Control Variate**). *Let $X$, $Y$ and $Z$ be random variables where the law of $X$ conditional on $Z$ is denoted $\mathbb{P}_{\boldsymbol{\theta}}(X|Z)$, and $Y$ is independent of $X$ conditioned on $Z$; i.e. $X \perp\!\!\!\perp Y \mid Z$. Then, we have that $\mathbb{E}[Y \nabla_{\boldsymbol{\theta}} \ln \mathbb{P}_{\boldsymbol{\theta}}(X)] = 0$.*

*Proof.* The proof follows from the law of iterated expectations:

$$\mathbb{E}[Y \nabla_{\boldsymbol{\theta}} \ln \mathbb{P}_{\boldsymbol{\theta}}(X)] = \mathbb{E}[\mathbb{E}[Y \nabla_{\boldsymbol{\theta}} \ln \mathbb{P}_{\boldsymbol{\theta}}(X) \mid Z]] = \mathbb{E}[\mathbb{E}[Y \mid Z] \mathbb{E}[\nabla_{\boldsymbol{\theta}} \ln \mathbb{P}_{\boldsymbol{\theta}}(X) \mid Z]] = 0,$$

since $\mathbb{E}[\nabla_{\boldsymbol{\theta}} \ln \mathbb{P}_{\boldsymbol{\theta}}(X) \mid Z] = 0$. $\blacksquare$

**Lemma 4.1.** *Factor baselines are valid control variates if $\mathcal{G}_{\Sigma}$ is true to the MDP (i.e. unbiased).*

*Proof.* Consider an objective $J(\boldsymbol{\theta})$ of the form defined in Equation 3, a factored influence network $\mathcal{G}_{\Sigma}$ and a $\Sigma$-factored policy $\pi_{\boldsymbol{\theta}}(\boldsymbol{a} \mid s) \doteq \prod_{i=1}^{n} \pi_{i,\boldsymbol{\theta}}(\sigma_i^{\pi}(\boldsymbol{a}) \mid s)$. Now, let us define a stochastic policy gradient estimator

$$\nabla_{\boldsymbol{\theta}} J(\boldsymbol{\theta}) = \mathbb{E}_{\pi_{\boldsymbol{\theta}}, \rho_{\pi_{\boldsymbol{\theta}}}} \left[ \boldsymbol{g}(s, \boldsymbol{a}) \doteq \sum_{i=1}^{n} \left[ \psi(s, \boldsymbol{a}) + b_i^{\mathrm{C}}(s, \bar{\sigma}_i^{\pi}(\boldsymbol{a})) \right] \boldsymbol{z}_i \right],$$

where $\boldsymbol{z}_i \doteq \nabla_{\boldsymbol{\theta}} \ln \pi_{i,\boldsymbol{\theta}}(\sigma_i^{\pi}(\boldsymbol{a}) \mid s)$ and $b_i^{\mathrm{C}}(s, \bar{\sigma}_i^{\pi}(\boldsymbol{a}))$ is the $i^{\mathrm{th}}$ factor baseline (see Definition 4.1). If $\mathcal{G}_{\Sigma}$ is unbiased then we have mutual independence between each action partition and, since $b_i^{\mathrm{C}}(s, \bar{\sigma}_i^{\pi}(\boldsymbol{a}))$ depends only on $s$ and $\bar{\sigma}_i^{\pi}(\boldsymbol{a})$ — i.e. the action elements that are not in the support of $\pi_{i,\boldsymbol{\theta}}$ — we can readily apply Lemma A.1, thus concluding the proof. $\blacksquare$

### A.1 Optimality

In contrast to the factor baselines, solving for the optimal baseline in general is a non-trivial challenge. Indeed, the results presented by Wu et al. [57] rely on a key assumption that the policy factors do not share parameters in order to simplify the analysis; i.e. that $\langle \boldsymbol{z}_i, \boldsymbol{z}_j \rangle \approx 0$ for any $i, j \in [|\Sigma|]$. Below we explain why this is a difficult problem, and leave it to future work to find the solution.

For notational convenience, let $\boldsymbol{g}(s, \boldsymbol{a}) \doteq \sum_{i=1}^{n} \boldsymbol{g}_i(s, \boldsymbol{a})$ such that the total variance on the gradient is given by

$$\mathbb{V}[\boldsymbol{g}(s, \boldsymbol{a})] = \sum_{i=1}^{n} \sum_{j=1}^{n} \mathrm{Cov}[\boldsymbol{g}_i(s, \boldsymbol{a}), \boldsymbol{g}_j(s, \boldsymbol{a})]. \tag{9}$$

The $n$ *optimal baselines* are given by the values that minimise Equation 9; i.e. $b_i^{\star}(s, \bar{\sigma}_i^{\pi}(\boldsymbol{a})) \doteq \arg\min_{b_i} \mathbb{V}[\boldsymbol{g}(s, \boldsymbol{a})]$ for all $i \in [n]$. To solve this problem, we first apply the factor baseline decomposition such that $b_i^{\star}(s, \bar{\sigma}_i^{\pi}(\boldsymbol{a})) = b_i^{\mathrm{V}}(s, \bar{\sigma}_i^{\pi}(\boldsymbol{a})) + b_i^{\mathrm{C}}(s, \bar{\sigma}_i^{\pi}(\boldsymbol{a}))$. This implies that the optimisation problem can be reduced to finding $\arg\min_{b_i^{\mathrm{V}}} \mathbb{V}[\boldsymbol{g}(s, \boldsymbol{a})]$ when $b_i$ is replaced with $b_i^{\star}$ for all $i \in [n]$. Now, let $\boldsymbol{x}_i \doteq [\boldsymbol{K}_{\Sigma} \psi(s, \boldsymbol{a})]_i \boldsymbol{z}_i$ and $\boldsymbol{y}_i \doteq b_i^{\mathrm{V}}(s, \bar{\sigma}_i^{\pi}(\boldsymbol{a})) \boldsymbol{z}_i$ such that $\boldsymbol{g}_i(s, \boldsymbol{a}) = \boldsymbol{x}_i + \boldsymbol{y}_i$. Note that while $\boldsymbol{y}_i$ depends on the full action, $\boldsymbol{x}_i$ depends only on the actions influencing the targets in $[\boldsymbol{K}_{\Sigma} \psi(s, \boldsymbol{a})]_i$. Removing terms that are independent of $b_i^{\mathrm{V}}$ thus yields the following:

$$\arg\min_{b_i^{\mathrm{V}}} \mathbb{V}[\boldsymbol{g}] = \arg\min_{b_i^{\mathrm{V}}} \left\{ \mathbb{V}[\boldsymbol{g}_i] + \sum_{j \neq i}^{n} \mathrm{Cov}[\boldsymbol{g}_i, \boldsymbol{g}_j] \right\},$$

$$= \arg\min_{b_i^{\mathrm{V}}} \left\{ \mathbb{V}[\boldsymbol{x}_i] + \mathbb{V}[\boldsymbol{y}_i] + 2\,\mathrm{Cov}[\boldsymbol{x}_i, \boldsymbol{y}_i] + \sum_{j \neq i}^{n} \mathrm{Cov}[\boldsymbol{x}_i, \boldsymbol{x}_j] + \mathrm{Cov}[\boldsymbol{x}_i, \boldsymbol{y}_j] + \mathrm{Cov}[\boldsymbol{y}_i, \boldsymbol{x}_j] + \mathrm{Cov}[\boldsymbol{y}_i, \boldsymbol{y}_j] \right\}$$

$$= \arg\min_{b_i^{\mathrm{V}}} \left\{ \mathbb{V}[\boldsymbol{y}_i] + 2\,\mathrm{Cov}[\boldsymbol{x}_i, \boldsymbol{y}_i] + \sum_{j \neq i}^{n} \mathrm{Cov}[\boldsymbol{x}_i, \boldsymbol{y}_j] + \mathrm{Cov}[\boldsymbol{y}_i, \boldsymbol{x}_j] + \mathrm{Cov}[\boldsymbol{y}_i, \boldsymbol{y}_j] \right\}.$$

To solve the equation above, we first expand each component and remove any redundant terms. For the variance on $y_i$, we have that

$$\mathbb{V}[y_i] = \mathbb{E}_a\left[(b_i^{\mathsf{V}})^2 \langle z_i, z_i \rangle\right] + \langle \mathbb{E}_a[b_i^{\mathsf{V}} z_i], \mathbb{E}_a[b_i^{\mathsf{V}} z_i] \rangle,$$

$$= \mathbb{E}_a\left[(b_i^{\mathsf{V}})^2 \langle z_i, z_i \rangle\right] + \langle \mathbb{E}_{\bar{\sigma}_i^\pi(a)}[b_i^{\mathsf{V}}] \mathbb{E}_{\sigma_i^\pi(a)}[z_i], \mathbb{E}_{\bar{\sigma}_i^\pi(a)}[b_i^{\mathsf{V}}], \mathbb{E}_{\sigma_i^\pi(a)}[z_i] \rangle,$$

$$= \mathbb{E}_a\left[(b_i^{\mathsf{V}})^2 \langle z_i, z_i \rangle\right],$$

$$= \mathbb{E}_{\sigma_i^\pi(a)}[\langle z_i, z_i \rangle] \, \mathbb{E}_{\bar{\sigma}_i^\pi(a)}\left[(b_i^{\mathsf{V}})^2\right]. \tag{10}$$

It follows from this analysis that the covariance between $y_i$ and $y_j$ for any $i, j \in [|\Sigma|]$, with $i \neq j$, is given by

$$\mathrm{Cov}[y_i, y_j] = \mathbb{E}_a\left[b_i^{\mathsf{V}} b_j^{\mathsf{V}} \langle z_i, z_j \rangle\right]. \tag{11}$$

Finally, we can expand the covariance between $x_i$ and $y_i$,

$$\mathrm{Cov}[x_i, y_i] = \mathbb{E}_a\left[[K_\Sigma \psi]_i \, b_i^{\mathsf{V}} \langle z_i, z_i \rangle\right] + \langle \mathbb{E}_a[[K_\Sigma \psi]_i \, z_i], \mathbb{E}_a[b_i^{\mathsf{V}} z_i] \rangle,$$

$$= \mathbb{E}_a\left[[K_\Sigma \psi]_i \, b_i^{\mathsf{V}} \langle z_i, z_i \rangle\right] + \langle \mathbb{E}_a[[K_\Sigma \psi]_i \, z_i], \mathbb{E}_{\bar{\sigma}_i^\pi(a)}[b_i^{\mathsf{V}}] \mathbb{E}_{\sigma_i^\pi(a)}[z_i] \rangle,$$

$$= \mathbb{E}_a\left[[K_\Sigma \psi]_i \, b_i^{\mathsf{V}} \langle z_i, z_i \rangle\right],$$

$$= \mathbb{E}_{\bar{\sigma}_i^\pi(a)}[b_i^{\mathsf{V}}] \, \mathbb{E}_{\sigma_i^\pi(a)}[[K_\Sigma \psi]_i \langle z_i, z_i \rangle], \tag{12}$$

and similarly resolve the cross-covariance terms:

$$\mathrm{Cov}[x_i, y_j] = \langle \mathbb{E}_{\bar{\sigma}_i^\pi(a)}[b_i^{\mathsf{V}} z_i], \mathbb{E}_{\sigma_i^\pi(a)}[[K_\Sigma \psi]_i \, z_j] \rangle. \tag{13}$$

The quantities above provide us with a platform to find solutions. For example, the optimal baseline approximation proposed by Wu et al. [57] can be found if we assume that $\langle z_i, z_j \rangle \approx 0$ since Equation 11 and Equation 12 go to zero. However, in the general case the problem is not quite so simple. The reason for this is that the baselines interact via the cross-covariance term in Equation 11. As a result, we cannot solve for each $b_i^{\mathsf{V}}$ independently of the others. Instead, we have a system of polynomial equations which may not have a unique solution. In fact, since each equation has degree $d = 2$, it follows the number of solutions can be as large $d^{|\Sigma|}$. In general, there are very few methods that can solve these type of systems, and those that can are limited to bounds of approximately $d^{|\Sigma|} \approx 20$. It seems reasonable to assume that any solution, while viable, would be computationally impractical, but we leave it to future work to establish this result formally.

## B  Factored Policy Gradients

The validity of factor baselines, as shown in the previous section, extends to policy gradient themselves. As discussed in the main text, we can show that FPGs are unbiased and satisfy certain variance bounds compared with conventional policy gradients. We restate the original propositions below and provide the proofs in full.

**Proposition 1.** *Take a $\Sigma$-factored policy $\pi_\theta(a|s)$ and $|\theta| \times |\Sigma|$ matrix of scores $S(s, a)$. Then, for target vector $\psi(s, a)$ and multipliers $\lambda$, the* FPG *estimator*

$$g^{\mathrm{C}}(s, a) \doteq S(s, a) \, K_\Sigma \, \lambda \circ \psi(s, a),$$

*is an unbiased estimator of the true policy gradient; i.e. $\nabla_\theta J(\theta) = \mathbb{E}_{\pi_\theta, \rho_{\pi_\theta}}[g^{\mathrm{C}}(s, a)]$.*

*Proof.* Let $\mathcal{G}_\Sigma$ denote an $\Sigma$-factored influence network with policy $\pi_\theta(a \mid s) \doteq \prod_{i=1}^n \pi_{i,\theta}(\sigma_i^\pi(a) \mid s)$, and global target function $\psi(s, a) = \sum_{j=1}^m \lambda_j \psi_j(s, \sigma_j(a)) = \langle \lambda, \psi(s, a) \rangle$. The score matrix, $S(s, a) \doteq \left[z_1^\top, \ldots, z_n^\top\right]^\top$, then has size $|\theta| \times n$, where $z_i \doteq \nabla_\theta \ln \pi_{i,\theta}(\sigma_i^\pi(a) \mid s)$. From this we can express the conventional policy gradient with no baseline as the linear product $g(s, a) = S(s, a) \, J_{n,m} \psi(s, a)$, where $J_{n,m}$ is the $n \times m$ all-ones matrix. By Lemma 4.1 the factor baselines, $[(1 - K_\Sigma) \psi(s, a)]_i$, are valid control variates and thus have expected value of zero under $\pi$. This means that they can be subtracted without introducing bias in the policy gradient, yielding

$$g(s, a) = \underbrace{S(s, a) \, J_{n,m} \psi(s, a)}_{\text{Vanilla PG}} - \underbrace{S(s, a) \, (1 - K_\Sigma) \, \psi(s, a)}_{\text{Factor Correction}} = S(s, a) \, K_\Sigma \, \psi(s, a).$$

It follows that $\nabla_{\boldsymbol{\theta}} J(\boldsymbol{\theta}) = \mathbb{E}_{\pi_{\boldsymbol{\theta}}, \rho_{\pi_{\boldsymbol{\theta}}}}\big[\boldsymbol{g}^{\mathrm{C}}(s, \boldsymbol{a})\big]$ since $\mathbb{E}_{\pi_{\boldsymbol{\theta}}, \rho_{\pi_{\boldsymbol{\theta}}}}\big[\boldsymbol{g}^{\mathrm{V}}(s, \boldsymbol{a})\big] = \mathbb{E}_{\pi_{\boldsymbol{\theta}}, \rho_{\pi_{\boldsymbol{\theta}}}}\big[\boldsymbol{g}^{\mathrm{C}}(s, \boldsymbol{a})\big]$ which concludes the proof. ∎

**Proposition 2.** *Let $\boldsymbol{g}_i$ denote a gradient estimate for the $i^{\text{th}}$ factor of a $\Sigma$-factored policy $\pi_{\boldsymbol{\theta}}$ (Equation 5). Then, $\Delta \mathbb{V}_i \doteq \mathbb{V}\big[\boldsymbol{g}_i^{\mathrm{V}}\big] - \mathbb{V}\big[\boldsymbol{g}_i^{\mathrm{C}}\big]$, satisfies*

$$\Delta \mathbb{V}_i = \alpha_i \, \mathbb{E}_{\bar{\sigma}_i^{\pi}(\boldsymbol{a})}\Big[\big(b_i^{\mathrm{C}}\big)^2\Big] + 2\beta_i \mathbb{E}_{\bar{\sigma}_i^{\pi}(\boldsymbol{a})}\big[b_i^{\mathrm{C}}\big],$$

*where $\boldsymbol{z}_i \doteq \nabla_{\boldsymbol{\theta}} \ln \pi_{i,\boldsymbol{\theta}}(\boldsymbol{a} \,|\, s)$, $\alpha_i \doteq \mathbb{E}_{\sigma_i^{\pi}(\boldsymbol{a})}[\langle \boldsymbol{z}_i, \boldsymbol{z}_i \rangle] \geq 0$ and $\beta_i \doteq \mathbb{E}_{\sigma_i^{\pi}(\boldsymbol{a})}\big[\langle \boldsymbol{z}_i, \boldsymbol{z}_i \rangle \big(\psi + b_i^{\mathrm{C}}\big)\big]$.*

*Proof.* First, let us denote by $\boldsymbol{X}$ and $\boldsymbol{Y}$ two (possibly dependent) random variables, with $\boldsymbol{Z} \doteq \boldsymbol{X} - \boldsymbol{Y}$ such that

$$\begin{aligned}
\Delta \mathbb{V} \doteq \mathbb{V}[\boldsymbol{X}] - \mathbb{V}[\boldsymbol{Y}] &= \mathbb{V}[\boldsymbol{Z} + \boldsymbol{Y}] - \mathbb{V}[\boldsymbol{Y}], \\
&= \mathbb{V}[\boldsymbol{Z}] + \mathbb{V}[\boldsymbol{Y}] + 2\mathrm{Cov}[\boldsymbol{Y}, \boldsymbol{Z}] - \mathbb{V}[\boldsymbol{Y}], \\
&= \mathbb{V}[\boldsymbol{Z}] + 2\,\mathrm{Cov}[\boldsymbol{Y}, \boldsymbol{Z}].
\end{aligned}$$

From Proposition 1, we can express the vanilla and factored policy gradient estimators for the $i^{\text{th}}$ factor as $\psi\,\boldsymbol{S}_{\cdot,i}$ and $\big(\psi - b_i^{\mathrm{C}}\big)\boldsymbol{S}_{\cdot,i}$, respectively, where the function arguments have been omitted for clarity. Assigning these values to $\boldsymbol{X}$ and $\boldsymbol{Y}$ we arrive at the equality relations

$$\mathbb{V}[\boldsymbol{Z}] = \mathbb{V}\big[b_i^{\mathrm{C}} \boldsymbol{z}_i\big] = \mathbb{E}_\pi\Big[\langle \boldsymbol{z}_i, \boldsymbol{z}_i \rangle \big(b_i^{\mathrm{C}}\big)^2\Big]$$

$$\mathrm{Cov}[\boldsymbol{Y}, \boldsymbol{Z}] = \mathrm{Cov}\big[\big(\psi - b_i^{\mathrm{C}}\big)\boldsymbol{z}_i, b_i^{\mathrm{C}} \boldsymbol{z}_i\big] = \mathbb{E}_\pi\big[\langle \boldsymbol{z}_i, \boldsymbol{z}_i \rangle \big(\psi - b_i^{\mathrm{C}}\big) b_i^{\mathrm{C}}\big].$$

The former follows from the fact that $\mathbb{E}_\pi[\boldsymbol{S}_{\cdot,i}] = 0$ for all $i$, and latter by noting that $\mathbb{E}[\boldsymbol{Z}] = 0$ due to Lemma 4.1. We can now exploit the independencies implied by the influence network, $\mathcal{G}_\Sigma$, to give

$$\Delta \mathbb{V}_i = \mathbb{E}_{\sigma_i^{\pi}(\boldsymbol{a})}[\langle \boldsymbol{z}_i, \boldsymbol{z}_i \rangle]\,\mathbb{E}_{\bar{\sigma}_i^{\pi}(\boldsymbol{a})}\Big[\big(b_i^{\mathrm{C}}\big)^2\Big] + 2\mathbb{E}_{\sigma_i^{\pi}(\boldsymbol{a})}\big[\langle \boldsymbol{z}_i, \boldsymbol{z}_i \rangle \big(\psi - b_i^{\mathrm{C}}\big)\big]\,\mathbb{E}_{\bar{\sigma}_i^{\pi}(\boldsymbol{a})}\big[b_i^{\mathrm{C}}\big],$$

This is the desired result and thus concludes the proof. ∎

**Corollary 4.1.** *Let $\psi(s, \boldsymbol{a})$ be of the form in Equation 3. If $\psi_j(s, \boldsymbol{a}) \geq \underline{\psi_j}$ for all $(s, \boldsymbol{a}) \in \mathcal{S} \times \mathcal{A}$ and $j \in [m]$, with $\big|\underline{\psi_j}\big| < \infty$, then there exists a linear translation, $\psi_i \to \psi_i - \sum_{j=1}^m \lambda_j \underline{\psi_j}$, which leaves the gradient unbiased but yields $\Delta \mathbb{V}_i \geq 0$.*

*Proof.* Take a target set $\Psi$ and let $\underline{\psi_j} \doteq \inf_{\mathcal{S},\mathcal{A}} \psi_j$ for each $\psi \in \Psi$. The unbiasedness claim follows from the fact that these terms go to zero in expectation when weighted by the score functions; they are constants. The variance claim is also trivial, since $\psi_j + \sum_{k=1}^m \lambda_k \inf_{\mathcal{S},\mathcal{A}} \psi_k$ are non-negative and, due to the summation over all $k \in [m]$, no CB can yield a negative value. Each term in Equation 8 (Proposition 2) must also be non-negative, which concludes the proof. ∎

## C   Minimum Factorisation

The minimum factorisation of an influence network provides a natural way of partitioning action nodes into independent policy distributions. In the main text it was also stated that such a characterisation is natural to the problems we study. We repeat this result below and provide the proof herein.

**Theorem 4.1.** *The MF $\Sigma_{\mathcal{G}}^\star$ always exists and is unique.*

*Proof.* Bipartite graphs always have at least one valid biclique and thus MF. Now, for uniqueness, let $\mathcal{G}$ denote an influence network. If $\mathcal{G}$ is complete, then we automatically satisfy the uniqueness property since the MF will contain a single biclique that covers all vertices in $I_\mathcal{A}$. If $\mathcal{G}$ is incomplete, then the proof can be shown through contradiction. Suppose that $A$ and $B$ are both MFs and therefore correspond to minimum biclique vertex covers, disjoint amongst $I_\mathcal{A}$. We know then that $A$ and $B$ must have the same dimensionality since they are optimal — i.e. contain the same number of bicliques — but, if they are distinct, then there must also exist at least one biclique $a \in A$ that is not in $B$. Since both MFs are defined over the same graph $\mathcal{G}$, the elements of $a$ must be distributed between at least 2 distinct bicliques in $B$. However, if this is the case, the union of these subgraphs would also form a valid biclique. The new cover, $B'$, containing the merged bicliques is valid and has dimensionality $|B'| < |B| = |A|$. This implies that neither $A$ nor $B$ can be MFs. Since the same must be true for any $A$ and $B$, it follows that there can be only one MF, thus concluding the proof. ∎

## D   Search Bandit

The search bandit was designed to exhibit an influence network as illustrated in Figure 6. Below we summarise the hyperparameters for the two key experiments — namely the baseline comparison (BC) and aliasing demonstration (AD):

**BC**   All algorithms were trained using a learning rate of 0.5 except for VPGs w/o a baseline which was only stable with a step size of 0.001. The state-based (i.e. scalar) baselines, $b(s) = b$, were trained using temporal-difference methods with a learning rate of 0.1. The action-dependent baseline, $b(s, a) \doteq -||\boldsymbol{a} - \boldsymbol{w}||_1 / |\mathcal{A}|$, was similarly trained using SARSA with a learning rate of 0.1.[4] An additional 1000 episodes were also used at the start of each run to pre-train the baseline if used.

**AD**   In the aliasing experiment, both VPGs and FPGs were trained for a 100-dimensional action-space with a regularisation penalty of $\lambda = 0.01$ on the first 99 action-components. VPGs were instantiated with a learning rate of 0.001, and FPGs with a rate of 0.01.

**Additional results.**   In addition to the results presented in the paper, we also include Figures 7-9. These explore the impact of the factor baseline across a set of dimensionalities and learning rates. We show that VPGs are very sensitive to the learning rate, especially when $|\mathcal{A}|$ is large. FPGs, on the other hand, converge on the optimal solution consistently regardless of the problem instance. Similarly, we show that the mean number of steps required to reach such a solution for a finite budget is much lower for FPGs compared with VPGs.

**Implications for MDPs.**   The search bandit is an interesting problem environment because, in many ways, it can emulate the learning process in arbitrary MDPs. This follows because, without loss of generality, we can always transform an MDP into a (possibly infinite) set of continuum multi-armed bandits, one occupying every unique state $s \in \mathcal{S}$. The question is how to define the cost function in order to achieve some form of equivalence. For example, if we consider deterministic policies, then we can clearly define the cost to be $\text{Cost}(\boldsymbol{a}) \doteq ||\boldsymbol{a} - \pi^\star(s)||_p$, for $p \geq 1$, and have the same solution set as given under Bellman optimality. This implies that the performance observed in the search bandit it likely to tell us about the performance in full MDPs. The results presented in Section 5.1 may thus provide evidence that FPGs will outperform VPGs for arbitrarily challenging MDPs.

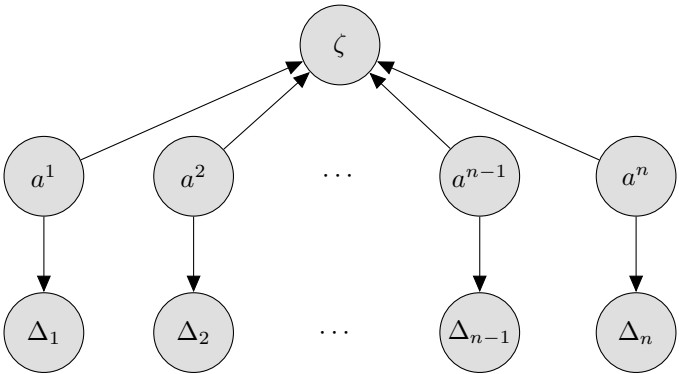

Figure 6: Influence network of the search bandit problem with optional coupling term.

## E   Traffic Systems

The traffic experiment were kept as close as possible to the benchmark specification for the grid problem provided by Flow [56]. In particular, we based the code of the "examples/exp_configs/rl/multiagent/multiagent_traffic_light_grid.py" and "examples/exp_configs/rl/signleagent/singleagent_traffic_light_grid.py" files on commit ID 4e47f7a. The

---

[4]In this formalism we only have sub-derivatives. For simplicity we simply assigned the gradient when a given action was equal to the weight.

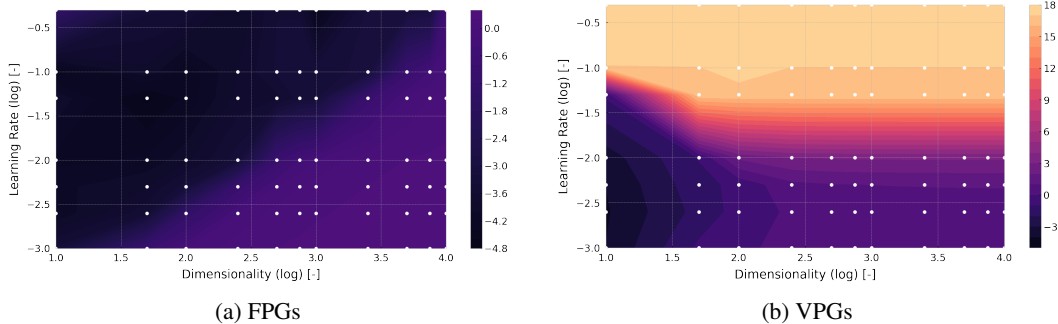

(a) FPGs

(b) VPGs

Figure 7: Mean optimality gap after $2 \times 10^5$ training iterations. The $z$-axis is given in a log scale and each point was computed from 16 random samples under the assumption of a Gamma distribution (optimality gap is lower bounded at zero).

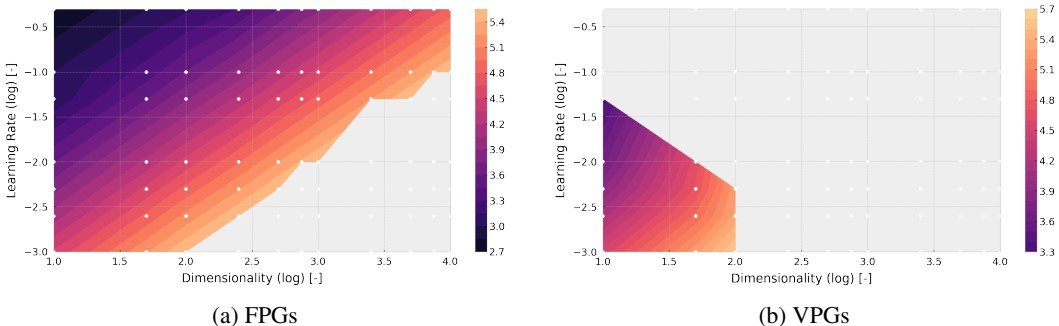

(a) FPGs

(b) VPGs

Figure 8: Mean number of time steps required to reach an optimality gap of $0.1$, up to a limit of $5 \times 10^5$ training iterations; see Figure 7. The $z$-axis is given in a log scale, and unfilled (grey) regions depict either divergence or a failure to terminate in the allotted time. Each point was computed from 16 random samples under the assumption of a Gamma distribution (time is lower bounded at zero).

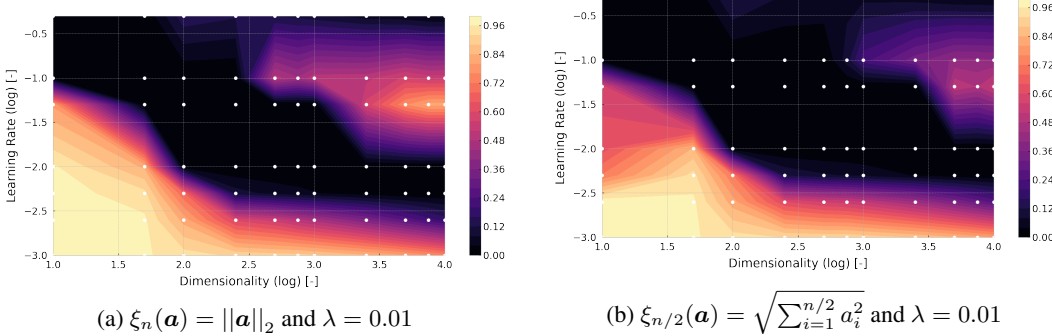

(a) $\xi_n(\boldsymbol{a}) = ||\boldsymbol{a}||_2$ and $\lambda = 0.01$

(b) $\xi_{n/2}(\boldsymbol{a}) = \sqrt{\sum_{i=1}^{n/2} a_i^2}$ and $\lambda = 0.01$

Figure 9: Ratio between the mean optimality gaps for FPGs over VPGs after $2 \times 10^5$ training iterations. Smaller values indicate that FPGs achieved a lower error relative to VPGs. Each point is the ratio of the two means, each computed using 16 random samples under the assumption of a Gamma distribution (optimality gaps are lower bounded at zero).

only changes that were made were to update the topology of the grid (i.e. $3 \times 3$ and $2 \times 6$), and to unify the reward function. We outline all the specific details below.

**Reward functions.** In order to unify the reward function across domains we implemented a custom variant of the "mean delay" case that worked for single- or multi-agent approaches. In particular, we changed the summation to only consider a subset of the edges in the network which allowed for localised computation. This can be done very easily in the Flow framework.

**Traffic system parameters.** The traffic intersection problem was instantiated with either a $3 \times 3$ topology, or a $2 \times 6$ topology, depending on the experiment. In all cases, an edge inflow of 300 was used, with initial speed of 30. The inner edges were given a length of 300, with the final edge in a route having length 100, and starting edge having length 300. Cars were created using the SimCarFollowingController, and SumoCarFollowingParams with a minimum gap of 2.5, maximum speed of 30, decelleration rate of 7.5 and "right of way" speed mode. The environment itself was initialised with target velocity of 50, switch time of 3, number of locally observed cars at 2, "actuated" TL type, and 4 locally observed edges.

**Learning hyperparameters.** In all cases we leveraged RLLIB's implementation of PPO with GAE [21] using discount factor of 0.999, a Monte-Carlo interpolation rate of $\lambda = 0.97$, KL-target of 0.02, value function clipping bound at $10^4$, and learning rate of $5 \times 10^{-4}$. The policy was parameterised using a three-layer neural network with 32 units at each of the three hidden layers. A total of 50 CPUs were used, each generating a single rollout at each iteration with a horizon of 400 steps.