# OpenReview forum: "Factored Policy Gradients: Leveraging Structure for Efficient Learning in MOMDPs"
_NeurIPS.cc/2021/Conference — NeurIPS 2021 Poster_

### Official Review · Reviewer_BRAD · 2021-07-13

**Rating:** 6
**Confidence:** 3

**Summary:**

This paper studies multi objective MDPs with a factorized action space and proposes a new “baseline” for policy gradient related actor-critic sorts of approaches for the class of problems that are of interest to them. They show variance reduction and perform experiments to gauge the power of the proposal.

**Ethical Concerns:**

I don't have any specific ethical concerns.

**Limitations And Societal Impact:**

The authors should clarify that their approach works for a specific sub-class of problems. Other than that, I don't see issues.

**Main Review:**

On the face of it, the paper makes contributions that seem quite reasonable to the actor-critic and policy gradient literature for a specific class of MDP problems: those involving factored action space and multiple objectives. I’m sure this class of problem is large enough such that the paper provides useful contributions to the literature. However, I do not have expertise in these subjects and will defer to my co-reviewers to gauge the work in this space.

My main concern with the paper is that it seems to have a major positioning problem. I have more expertise in graphical models and causality – and the problem with the paper’s positioning is probably why the paper was assigned to me. Unfortunately, the paper has very little to do with causality, even though the authors choose to mention causality to describe their factored model. There seems to be a trend these days where people add the word “causal” to terms. I find the phrase “causal policy gradients” to be unsuitable for their approach for many reasons, but primarily as it seems to imply that it applies to a broader class of problems than it really does. I strongly recommend changing the term and re-positioning the paper. The fact that problems can be factorized as described in the paper and that values could depend on different decision (action) variables is not at all surprising, and there is no need to mention causality (I mention some relevant literature on graphical models later). If the authors disagree, they should feel free to explain their choice of terminology.

Detailed comments:

P1: What are “daemon agents”?

P2: The word “baseline” is used in a few places on this page, but it is not until much later that it becomes clear what this refers to. I suggest the authors modify the introduction to mention actor-critic and other ideas earlier, so that the contributions make sense to a general AI reader.

P3: The description of factorization in the action space was not totally clear as described. What does it mean for actions to be “accessible” in an element-wise fashion? Does it simply mean that the action space is a cross-product of individual action spaces, and that they could individual affect different aspects of reward (like in Figure 1)? A real example would help here, and in other places.

P3: On line 126, psi_1 is mentioned to be equivalent to q – should this be v?

P4: The authors are perhaps not familiar with influence diagrams – these are graphical models for decision problems, including MDPs (including multi-objective one). Their influence network can be represented as an influence diagram, with arcs from decision/action variables to value nodes. The idea of factorization using additive value functions is well known in that literature. Here is an older paper primarily about dynamic problems, but the ideas extend to multi-objective problems too:
J. A. Tatman and R. D. Shachter (1990), Dynamic Programming and Influence Diagrams, IEEE Transactions on Systems, Man and Cybernetics 20(2):365-379.

P4: I’m surprised that actions are never allowed to depend on other actions in a policy. What is the reason for this simplification?

P4: The authors mention the market making problem in Section 3.1 as an application. Could they provide more detail? More motivation around the problem setting would help the paper greatly.

P4: It looks like there is a self-reference in Def. 3.3.

P4: “n” seems to be used both for the number of actions in the action space (earlier) but also for the cardinality of Sigma, which can probably be less than “n” in general. There is also an “m” mentioned earlier. Could the authors clarify please?

P5: As mentioned earlier, I find “causal policy gradients” to be an absurd term for the method. I suggest changing the title of Section 4. Perhaps “factored policy gradients” or something like that.

P6: I may have missed something but how can one tell the second term in Equation (8) is asymmetric. Is this inferred from the beta?

P7: There are terms in Def. 4.2 that were not defined. Also – I found most of Section 4.2 hard to parse. Whatever I did understand, I did not find surprising. I was not able to fully appreciate the implications of this section. From what I could tell, the main contributions of the paper are around the variance results in Section 4.1.

P7 and 8: Again, I did not appreciate Section 5.1. Perhaps this is because the example was abstract and did not give me a sense of the implications. How many objectives are involved in this synthetic example?

P8: Example 7 should probably be Example 4a.

P8: Why is “Flow” outstanding?

P9: I find line 355 (“one of the most promising directions …”) an overstatement and not at all backed by the paper. Please remove or rephrase that line.

**Time Spent Reviewing:**

4

---

> ### Author Response · Authors · 2021-08-09
> **Response to Reviewer BRAD**
>
> First of all, we would like to thank the reviewer for the time spent assessing our work. All feedback is valuable and we learn a great deal from your input.
>
> **Naming Issue**
>
> We would like to sincerely apologise to the reviewer, and more broadly, for the misuse of the term causal and for leading to an improper assignment. While we did not mean to "join in the hype," we appreciate that this was a disingenuous use of terminology. Our original motivation came from studying a real-world problem for which there is a concrete notion of causality with respect to the influence of different actions on the reward. However, we acknowledge that, while conceptually useful for understanding, it is not appropriate to use the term to describe the technique. We agree that a better name for the work (and thus positioning) is to call it "_Factored Policy Gradients_" as suggested. Should the paper be accepted we would change the name and remove headers/text that refer to notions causality.
>
> **Daemon Agents**
>
> Daemon agents are a colloquial term used by Sutton [1] to refer to sub-agents in an agent hierarchy. We will remove this and make things much clearer in the text by avoiding such jargon.
>
> **Baseline Definition**
>
> We agree that the term baseline is not well-defined in the introduction. We will address this in the text and ensure it is made clear to the reader, perhaps referring to the broader notion of control variates first and how they relate to the Monte-Carlo gradient estimation literature; e.g. [3].
>
> **Accessible Spaces**
>
> You are exactly correct - by accessible we mean that the action can be decomposed into its constituent elements; e.g. a tuple or vector. We do agree, however, that the explanation could be improved to make this clear using concrete examples.
>
> **Should $q$ be replaced with $v$**
>
> This assertion is not correct. If you were to assign $\psi$ the value $v$, it follows from the control variate lemma (see Lemma A.1 in the appendix) that the gradient would resolve to zero. One must always define a target that covarys with the policy factor distribution such that the expected value of the target-score product is not simply the product of their separate expectations. For more details on this, we refer the reviewer to Section 2 of [4] or Section 4 of [3].
>
> **Influence Diagrams**
>
> We thank the reviewer for the references. We were not aware of this particular line of work but it is clear that there is a strong connection with the ideas we are proposing: the equivalencies between their paper, our paper, and that of the influence-based abstraction literature [e.g. 5] (for which we do provide a discussion) are intriguing. In the next iteration of the paper we will include references to this line of work and discuss how our "influence network" relates to "influence diagrams". We strongly feel that this only adds further credibility to our approach since it borrows from what is clearly a very well-studied area. Furthermore, this connection could help open up a number of directions for future work, especially in combining our action-reward factorisation with the existing work on state-transition factorisation.
>
> **Actions Depending on Actions**
>
> The reviewer is correct that we do not consider the case of policy distributions that have a conditional structure with respect to other components of the same action. While this would be an interesting direction to explore, it is not something that is typically considered in RL; to the best of our knowledge. Indeed, relatively little work has been done on the impact of policy structure on performance of policy gradients. For the most part this direction has been limited to the study of topological properties of the action-space; see e.g. [6-8].
>
> Regardless, we do suspect that one could benefit from even further reductions in variance by accounting for the suggested structure. Indeed there are a number of possible directions one could explore in this area. However, it is not immediately clear how large the class of problems is that warrant such a representation, and we would leave this to future work to study properly.
>
> **Market Making**
>
> As the reviewer points out, the market making problem is a prime example of where this type of method would be very effective. Existing work in this area has also pointed out that scalability is an issue in RL for applications in this area [2]. We will expand on this suggestion in the paper, as we agree that the paper would greatly benefit from it, and summarise below:
>
> Market makers are liquidity providers in financial markets. They offer to buy and sell a given asset at all times such that there is always a counterparty to any transaction at some price. This is often a contractual obligation with an exchange. The key challenge for market makers is to balance the flow of transactions of the buy and sell sides so as not to expose themselves to large net positions in an asset; this has been studied extensively in math finance and RL. This problem is typically posed from a uni-asset, or uni-venue perspective such that there are only two controls (buy price and sell price). However, when one starts to consider more complex instances (i.e. multiple assets, multiple venues, or even over-the-counter markets) things get intractable very quickly. This is primarily caused by the rapid (quadratic) increase in variance due to exposure to increasing numbers of random variables (price processes and transaction dynamics) driving the reward. Our approach solves this problem entirely, and leverages the expertise available in the field of math finance and elsewhere to construct efficient attribution methods, rather than leave everything to learning.
>
> It is natural to imagine that other real-world problem settings would benefit similarly. Indeed, our approach would allow practitioners to scale RL methods to enormous action-spaces wherever prior knowledge of the structure of $r(s, a, s')$ is available.
>
> **Notational Issues**
>
> Yes, you are correct (both on the self-reference and reuse of $n$) and we will amend in the next update of the paper.
>
> **Asymmetric Term**
>
> The use of the word "asymmetric" was not intended in a precise sense. A better pair of terms would be "signed" and "unsigned" terms. We will make this change in the text, as we agree that use of "symmetry" is confusing to the reader.
>
> **Minimum Factorisation**
>
> In this section we try to address the immediate question that arises: "how does one factor a policy?" For large classes of MDPs, one can choose _any_ factorisation, and it may be unclear which one is most appropriate. Our characterisation of minimum factorisation tries to address this by suggesting that some structures in the influence network are more "natural" than others; and our uniqueness result simply validates it as a well defined concept. While we do agree with the reviewer that the text could be clearer (and we will address this), we argue that it is important to provide the reader with some intuition into this choice. While it may well be a standard topic in graphical modelling, it is not as common to discuss (policy) factorisation in the RL literature. If the results were not surprising to the reviewer, then we only see that as a strength.
>
> With regards to definitions - we will fix this in the updated version of the paper. The reviewer is absolutely correct that terminology such as "minimum biclique vertex cover" should be properly defined.
>
> **Bandit Problem**
>
> This problem is a pedagogical problem from the literature on baselines and policy gradient methods; see e.g. Section 5 of [9]. The setting is informative since it removes the complexity due to state-representation and allows us to study the impact of increasing $\left\lvert\mathcal{A}\right\rvert$ in isolation.
>
> The reviewer comments that the number of objectives is not well-defined. This is a mistake in the text, and we apologise for the confusion. Figure 4 was generated using an action-space of 1000 dimensions. This means that there were 1000 terms in the objective function. We will correct the text to amend this error, and hope that this comment has clarified things.
>
> **Remaining Comments**
>
> The remaining comments will all be addressed in the updated version of the paper. Note also that "Flow" is outstanding because it is a code library, that is all.
>
> **Bibliography**
>
> [1] Sutton, Richard S., et al. "Horde: A scalable real-time architecture for learning knowledge from unsupervised sensorimotor interaction." The 10th International Conference on Autonomous Agents and Multiagent Systems-Volume 2. 2011.
>
> [2] Guéant, Olivier, and Iuliia Manziuk. "Deep reinforcement learning for market making in corporate bonds: beating the curse of dimensionality." Applied Mathematical Finance 26.5 (2019): 387-452.
>
> [3] Mohamed, Shakir, et al. "Monte Carlo Gradient Estimation in Machine Learning." J. Mach. Learn. Res. 21.132 (2020): 1-62.
>
> [4] Schulman, John, et al. "High-dimensional continuous control using generalized advantage estimation." arXiv preprint arXiv:1506.02438 (2015).
>
> [5] Oliehoek, Frans, Stefan Witwicki, and Leslie Kaelbling. "Influence-based abstraction for multiagent systems." Proceedings of the AAAI Conference on Artificial Intelligence. Vol. 26. No. 1. 2012.
>
> [6] Chou, Po-Wei, Daniel Maturana, and Sebastian Scherer. "Improving stochastic policy gradients in continuous control with deep reinforcement learning using the beta distribution." International conference on machine learning. PMLR, 2017.
>
> [7] Eisenach, Carson, et al. "Marginal policy gradients: A unified family of estimators for bounded action spaces with applications." arXiv preprint arXiv:1806.05134 (2018).
>
> [8] Fujita, Yasuhiro, and Shin-ichi Maeda. "Clipped action policy gradient." International Conference on Machine Learning. PMLR, 2018.
>
> [9] Wu, Cathy, et al. "Variance reduction for policy gradient with action-dependent factorized baselines." arXiv preprint arXiv:1803.07246 (2018).

---

> > ### Comment · Reviewer_BRAD · 2021-09-02
> > **Thanks for the clarifications**
> >
> > I appreciate the response from the authors - particularly that they are willing to fix the positioning problem, which was one of my major concerns. I am increasing my score based on the response as well as discussions with other reviewers.

---

### Official Review · Reviewer_aQyY · 2021-07-16

**Rating:** 7
**Confidence:** 3

**Summary:**

This paper proposes a policy gradient algorithm for MOMDPs with large factored action spaces, where the background knowledge on the MOMDP structure is given as an influence network. A new baseline for policy gradients is given that makes use of the MOMDP background knowledge and a factorisation of the RL policy. The paper analyses the favourable variance properties of the resulting policy gradient estimator. Empirical results are provided on a pedagogical bandit example, and a traffic control problem. The latter demonstrates applicability of the method even when the graphical assumptions only hold approximately.

**Limitations And Societal Impact:**

I was satisfied with the discussion on societal impact in the broader impact statement. I felt that the limitations were transparently presented in the paper.

**Main Review:**

### Originality
+ The paper is the first I've seen to use causal background knowledge in this way to reduce the variance of policy gradient methods.
+ Graphical causal models have been used before in multi-agent problems (citation 27) but not for use in policy gradient methods on single-policy MOMDPs.

### Quality
+ I enjoyed the results of section 4.1 and the discussion of a bias-variance tradeoff in practice.
+ The experiments were thorough in the sense of studying properties of the approach beyond just performance. For example, the bias-variance tradeoff explored in figure 5b.
- In figure 5a, it does not seem meaningful to plot standard error bars for 5 repeats. Perhaps the min and max across repeats would be more informative. In 5b, with only 5 repeats one could just plot all of the repeats instead of giving the standard error bar.
- One of the main arguments for this approach in practice over using an action dependent baseline seems to be wall-clock time. Therefore I think including plots of this statistic in the appendix would be very valuable.

### Clarity
+ In general the writing is easy to follow and the figures ease the exposition well.
- Citation (56) computes the minimum variance baseline under the assumption that $\langle z_i, z_j \rangle =0$ (using the notation of the appendix). The paper nor appendix doesn't go far in explaining why in general settings we would not expect $\langle z_i, z_j \rangle =0$ to hold.
- I think when introducing the action-dependant baseline in the experiments a citation should be put after it for clarity (eg (56)).
- The paper describes the minimum factorisation approach as "principled" (in the main text) and "theoretically grounded" (in the appendix). It seems that this is done due to the existance and uniquenesss results. I therefore think the language used is slightly misleading, since it would imply to me that it is somehow the optimal factorisation (for, say, variance reduction) however there does not seem to be a result of this sort. Some discussion of the variance properties is given on line 263, but if there is a formal result regarding this, I think it should be written as such.
- Line 284 refers to figure 7. However, this is a figure in the appendix. The same is done with figure 9 on line 295.
- In line 322-324, it was clear to me what methods 1 and 3 were. However it was not clear what method 2 was. Was this the same as the action dependent baseline?
- Why did the action dependent baseline take two order of magnitude more wall-clock time? This was stated in section 5, but not explained. Was it due to fitting $\omega$ (notation of appendix) each time to compute the baseline?

### Significance
+ The method is simple, has nice theoretical results, and solid empirical performance. The use of background knowledge (influence graph) is reasonable and experiments show robustness to the assumptions not holding. Improving sample efficiency of policy gradients is certainly a direction many people are interested in.
- Traffic control problem is the only non-pedagogical example used for experiments. Demonstrating more general applicability would be nice. Perhaps in other cases it is not so easy to write down an approximate influence graph.
- The paper assumes that there is a scalarisable additive reward, which in MOMDPs is not true in general without prior knowledge.

**Time Spent Reviewing:**

7

---

> ### Author Response · Authors · 2021-08-05
> **Response to Reviewer aQyY**
>
> First of all, we would like to thank the reviewer for the time spent assessing our work. All feedback is valuable and we learn a great deal from your input.
>
> **Standard Errors in Figure 5**
>
> We agree with the reviewer that showing all bars in Figure 5b would be a better use of the results and we will update the plot in the next iteration of the paper. With regards to Figure 5a, we argue that even for 5 random samples, one can estimate a reasonable standard error of the mean. This issue of estimator accuracy is discussed in [1], and other works, who broadly find that a sample size of n ~ 5 or 6 leads to a small relative underestimation. We would argue that this is still informative compared to the max/min which may be very large in general. Indeed, standard errors and standard deviations are very common in the RL literature for measuring uncertainty and dispersion, respectively. We will, however, add a footnote to clarify this small underestimation concern.
>
> **Wall-Clock Times**
>
> We agree with the reviewer that illustrating the wall-clock performance more explicitly would be useful. We will add more detailed results of this to the appendix for the bandit setting should the paper be accepted, and add a more detailed explanation as to why the increase is observed for the action-dependent baseline. We also illustrate below the average rate at which updates were performed for the 5 methods in Figure. 4a (i.e. number of iterations per second - so higher is better). This will be added to the text to justify our comments in the text.
>
> | | Mean [it / s] | Standard Deviation [it / s] |
> | ----------- | ----------- | ----------- |
> | VPGs | 10534 | 87 |
> | VPGs (with $b(s)$) | 9885 | 81 |
> | VPGs (with $b(s, a)$) | 80 | 1 |
> | CPGs | 9950 | 157 |
> | CPGs (with $b(s)$) | 9670 | 126 |
>
> **Significance of $\langle \boldsymbol{z}_i, \boldsymbol{z}_j \rangle \approx 0$**
>
> As briefly mentioned in the appendix, this assumption (as made in [2]) essentially means that the parameters associated with the $i^{\textrm{th}}$ and $j^{\textrm{th}}$ policy factors are approximately independent (i.e. orthogonal); note that the score function of the policy will be a random variable due to the sampling of actions/transition dynamics. This does not hold in general. In particular, it does not hold when one _shares parameters across the policy factors_. For example, you could not use a single neural network with multiple heads, or simple a shared, conditional network. Instead, one would have to have $K$ seperate networks (where $K$ is the number of policy factors). This incurs a heavy cost on memory when the action-space is large (which goes against what we're trying to achieve), and reduces sample efficiency since we cannot share information across factors. Our proposed approach, in comparison, does not rely on this assumption for the analysis to hold.
>
> **Minimum Factorisation**
>
> As pointed out, the language used in the main text and appendix could be improved. We will tighten this up in the next version and ensure that there can be no ambiguity that we provide no optimality guarantees.
>
> **Method 2 on the Traffic Domain**
>
> To clarify, the second method we use is a factored policy (where each policy factor corresponds to one traffic light), but still using the total reward of the system (i.e. the performance for the entire network) to train the network. Note that, as with Method 3, we use a shared policy network which would break the $\langle \boldsymbol{z}_i, \boldsymbol{z}_j \rangle \approx 0$ of previous works. Namely, we train a singly policy network and condition on the identity of the traffic light we're operating at a given time.
>
> **Compute of Action-Dependent Baseline**
>
> You are exactly correct. The additional overhead in wall-clock time is due to learning the parameters, $\boldsymbol{\omega}$, of the baseline $b_{\boldsymbol{\omega}}(s, a)$. The reason for this is that the feature vector associated with the linear representation becomes very large as $\lvert\mathcal{A}\rvert$ grows. This incurs a heavy compute burden in comparison to our approach that leverages prior knowledge.
>
> **Bibliography**
>
> [1] Gurland, John, and Ram C. Tripathi. "A simple approximation for unbiased estimation of the standard deviation." The American Statistician 25.4 (1971): 30-32.
>
> [2] Wu, Cathy, et al. "Variance reduction for policy gradient with action-dependent factorized baselines." arXiv preprint arXiv:1803.07246 (2018).

---

> > ### Comment · Reviewer_aQyY · 2021-08-26
> > **response**
> >
> > Thank you to the authors for their response addressing my questions.

---

### Official Review · Reviewer_VuPN · 2021-07-17

**Rating:** 7
**Confidence:** 2

**Summary:**

This paper studies the problem of policy learning in a factored Markov decision process (MDP). The factorization of the transition probability distribution is graphically described through a set of influence diagrams. The objective function is a linear combination of weighted kernel functions. The authors propose an efficient policy gradient algorithm to learn the optimal policy. Simulation results support the authors’ results.

**Limitations And Societal Impact:**

- This work is mainly theoretical. Its long-term societal impact is unclear to see.

**Main Review:**

This paper considers an interesting problem in policy learning factored MDPs. I haven’t checked the details of the proof, but the proposed algorithm seems reasonable. Since many decision-making problems in the real world where its system dynamics can be naturally decomposed into a set of independent components, the method presented here could have many practical applications across disciplines.

In simulations, the authors mainly compare against a vanilla policy gradient that does not exploit the underlying factored representation. I am curious how the authors’ method compares against other policy learning algorithms for factored MDPs, e.g., (Guestrin et al., 2003, 2004). Could the authors further elaborate on this?


**Time Spent Reviewing:**

2 hours

---

> ### Author Response · Authors · 2021-08-05
> **Response to Reviewer VuPN**
>
> First of all, we would like to thank the reviewer for the time spent assessing our work. All feedback is valuable and we learn a great deal from your input.
>
> **Benchmark Algorithms**
>
> You are absolutely correct that there have been many other works in the past that investigate how to leverage structure in MDPs for more efficient learning. Indeed, one of the leading paradigms in theoretical research in RL is the class of "linear MDPs" [1, 2], and factored representations are also very common in multi-agent RL [3]. However, in the vast majority of these works, the emphasis is on _factorisation of transition dynamics_. This is a very different type of factorisation to what we study: namely, the influence of actions on the components of a linear-additive objective/reward function.
>
> The papers mentioned by Guestrin et al. [e.g. 4] fit into the past work in the sense that they also focus on transition dynamics. In particular, they propose to represent the dynamics function $p(s' | s, a)$ using a Dynamic Bayesian Network which allows for much more compact representations of state for function approximation and value estimation. As such, we do not feel like this would be an appropriate comparison for the methods discussed in our paper; for example, the bandit problem doesn't even have state. Instead, we include the state-of-the-art method of action-dependent baselines for reducing variance in policy gradient methods.
>
> To the best of our knowledge, our paper is indeed the first to explore this specific direction of incorporating action-reward dependence structure directly into a policy gradient estimator.
>
> **Bibliography**
>
> [1] Todorov, Emanuel. "Linearly-solvable Markov decision problems." Advances in neural information processing systems. 2007.
>
> [2] Neu, Gergely, and Julia Olkhovskaya. "Online learning in MDPs with linear function approximation and bandit feedback." arXiv preprint arXiv:2007.01612 (2020).
>
> [3] Oliehoek, Frans A., and Christopher Amato. A concise introduction to decentralized POMDPs. Springer, 2016.
>
> [4] Guestrin, Carlos, et al. "Efficient solution algorithms for factored MDPs." Journal of Artificial Intelligence Research 19 (2003): 399-468.

---

### Decision · Program_Chairs · 2021-09-27

**Decision:**

Accept (Poster)

**Comment:**

This paper introduces a novel causal baseline to reduce the variance in policy gradient methods. All reviewers like the paper and I recommend acceptance.